# IoT and Deep Learning-Based Farmer Safety System

**DOI:** 10.3390/s23062951

**Published:** 2023-03-08

**Authors:** Yudhi Adhitya, Grathya Sri Mulyani, Mario Köppen, Jenq-Shiou Leu

**Affiliations:** 1Department of Computer Science and Systems Engineering (CSSE), Graduate School of Computer Science and Systems Engineering, Kyushu Institute of Technology, 680-4 Kawazu, Iizuka-shi 820-8502, Fukuoka, Japan; 2Department of Electronic and Computer Engineering(ECE), National Taiwan University of Science and Technology, Taipei City 106, Taiwan

**Keywords:** time series dataset, quaternion, hierarchical temporal memory, cascade classifier, probability prediction, farming activity monitoring, smart farming

## Abstract

Farming is a fundamental factor driving economic development in most regions of the world. As in agricultural activity, labor has always been hazardous and can result in injury or even death. This perception encourages farmers to use proper tools, receive training, and work in a safe environment. With the wearable device as an Internet of Things (IoT) subsystem, the device can read sensor data as well as compute and send information. We investigated the validation and simulation dataset to determine whether accidents occurred with farmers by applying the Hierarchical Temporal Memory (HTM) classifier with each dataset input from the quaternion feature that represents 3D rotation. The performance metrics analysis showed a significant 88.00% accuracy, precision of 0.99, recall of 0.04, F_Score of 0.09, average Mean Square Error (MSE) of 5.10, Mean Absolute Error (MAE) of 0.19, and a Root Mean Squared Error (RMSE) of 1.51 for the validation dataset, 54.00% accuracy, precision of 0.97, recall of 0.50, F_Score of 0.66, MSE = 0.06, MAE = 3.24, and = 1.51 for the Farming-Pack motion capture (mocap) dataset. The computational framework with wearable device technology connected to ubiquitous systems, as well as statistical results, demonstrate that our proposed method is feasible and effective in solving the problem’s constraints in a time series dataset that is acceptable and usable in a real rural farming environment for optimal solutions.

## 1. Introduction

Within the past two decades, the Internet of Things (IoT) concept has evolved. IoT-connected devices can exchange information, commands, and decisions independently through intelligent networks within their ecosystem. The rapid development of IoT is opening up exceptional opportunities for various implementation sectors, including agriculture, healthcare, and manufacturing [1]. Implementing IoT infrastructure is expected to improve efficiency, reduce costs, and save energy, as well as prevent loss and accidents and open up new business opportunities.

Agriculture sectors frequently disregarded business cases for IoT implementation as it requires adapting to well-known farming activity [2,3,4]. Nevertheless, due to the remoteness of farming operations, IoT implementation proposes the possibility of transforming farmers’ practices. Even though this concept has yet to gain widespread acceptance, it is one of the IoT-based applications that must be acknowledged. Smart farming will become a significant application field in agriculturally-based countries [5,6,7]. Smart farming involves applying data and information technology in a complex farming system to be maximized. Data, information, and communication technologies are implemented to utilize machinery, equipment, and installation of sensors used in agricultural production systems. By leveraging the above-mentioned technologies, IoT [8] and cloud computing can strengthen the effectiveness and efficiency even further, one with innumerable sophisticated sensors, including the implementation of artificial intelligence into agriculture [9,10,11,12].

Work involving the management of livestock, tractors, and heavy agricultural equipment, exposure to toxic chemicals, or laboring in open fields under the sun’s heat or in a closed chamber, all have significant risks [13,14,15]. Employment in agriculture requires farm workers to work at elevations to finish their jobs. Falls from these elevations are the most expected agricultural casualties. Falling from a certain altitude can cause fractures, brain damage, and additional severe injuries. Falls, as the leading causes of unintentional injuries worldwide, with 37.3 million cases of falls each year, require medical attention [16].

Wearable technology appliances that use accelerometers and gyroscopes datasets have developed ecosystems and consumers [12,17,18,19,20]. Advanced technology gadgets that are linked to specific devices have become ubiquitous in our daily lives, ushering in a new trend in the mobile industry. These devices enable data generation, enabling the collection of massive quantities of data and information. Three angles can represent the 3D rotation [21,22] in studies using the Euler angle, which decomposes the rotation within three separate angles. Although using Euler angles is a natural step in the process of representing 3D rotations [23,24], it can cause difficulties in performing calculations, such as rounding errors in the conversion results, which can lead to distortion. More importantly, the Euler angle representation needs to be more linear. Farmers are recognizing the benefits of advancements in wearable technology, which have promising applications in healthcare products, monitoring, and personal health. Wearable technology can also be utilized in agriculture [25,26,27] and is expected to prepare preventative measurements for diverse categories of casualties in agriculture.

Deep learning can take advantage of IoT technology implementation. IoT sensors that can provide the input data of farmers’ positions and heart rates can be analyzed utilizing intelligent algorithms, combining data collected with previous data and training models to identify patterns and make predictions about the safety conditions of farmers in working environments, as depicted in Figure 1.

To overcome this problem, we proposed a quaternion to represent 3D rotation [28] as an input feature. While most of the current literature studies use three-dimensional data from accelerometer data and gyroscope data for activity prediction, we exploited quaternion as four-dimensional data (as a feed input for the Hierarchical Temporal Memory (HTM) classifier). We evaluated the approaches of large-scale egocentric datasets and farming pack mocap datasets; our proposed method is practical for computational efficiency and results. As a result, the primary goal of this study was to evaluate extracted quaternion features for hierarchical temporal memory input for accurate activity prediction.

The general framework for farmer safety systems in the smart farming environment is proposed in Figure 1. The farmer’s use of wearable sensor devices as an IoT subsystem has the potential to improve data readability in today’s medical systems, which can provide substantial sensor tracking of multiple physiological functions within the body, as well as cost-effective and high-efficiency services in a wide range of fields. These wearable devices generally use the battery as their power source and implement various power management strategies to extend battery life [29]. Most wearable devices can connect to the internet via Wi-Fi, and some even have their own cellular services. Accelerometer data and gyroscope data are collected on the device for local processing and transferred to the cloud by using the long-range (LoRa) network for decisions. This type of network has a distance range that can cover rural areas from the farm itself. The output of this computational process is then sent to the paramedics at the hospital. With these details, the caregiver can determine the proper action for preventing and following medical treatment measures [30,31,32] on what actions to take next; for example, sending an ambulance to the farmer, sending information to the farmer’s family about what happened to the farmer, and applying medical procedures at the hospital.

The goal is to exploit quaternion as four-dimensional data as the feed input for the classifier and compare the proposed algorithm to the other algorithm based on their strengths and vulnerabilities. In accordance with the findings from this study, the proposed framework significantly outperformed the others. The contributions of the proposed work are as follows:We utilized a quaternion as an input feature to represent 3D rotation.We evaluated the approaches on both large-scale egocentric datasets and farming pack mocap datasets, and demonstrate that our proposed method is practical regarding computational efficiency and result.Using the proposed algorithm, we determined the optimal solutions for the proposed system.With the same train test data, the proposed algorithm, HTM classifier, and (k-Nearest Neighbor) kNN classifier were compared.The performances of the two algorithms were compared using different performance metrics.

The rest of this paper is structured as follows: as this study is an IoT and deep learning-based farmer safety system, the related research on the farming activity monitoring method and the smart farming application include work on human motion classification and hierarchical classification algorithms, as introduced in Section 2. Section 3 presents the problem definition and clarifies the challenges behind the hierarchical temporal classification problem and its specificity. In Section 4, the material is proposed, i.e., in Section 4.1. Section 4.2, the proposed method and our proposed framework are formulated. The experimental results are presented in Section 5. Section 6 presents the discussion, and several studies that were compared to the existing ones are reported with numerical computations. Section 7 discusses future research directions and draws several conclusions.

## 2. Related Works

Health is a crucial priority [33], despite being the top revenue earner for numerous countries. Each country is exploring innovative practices to deliver quality healthcare [34,35]. The use of technology, particularly the internet, is essential to manage solutions effectively and efficiently. One such technology is machine learning algorithms combined with wearable healthcare devices [36]. Data and medical records can be collected to form a model that could be earlier by applying a machine learning algorithm [37,38].

Xiao Guo et al. [39] researched human motion prediction. The applied Skeleton Network (SkelNet) for local structure representation on different body components was used as input. The Skeleton Temporal Network (Skel-TNet) and Recurrent Neural Network (RNN) were used o apply the learning spatial and temporal dependencies. The final result was with the Residual Recurrent Neural Network (RRNN) and Convolutional Sequence to Sequence (CSS). Regardless, the analysis primarily focused on predicting short-term human motion.

The study by Judith Bütepage et al. [40] focuses on the use of the Deep Learning (DL) framework architecture to analyze human motion capture data. The study comprehends a generic representation and generalizes the motion capture data from generative models of 3D human motion. These feature representations of human motion are extracted via an encoding–decoding network that quantifies learned features from the current history, representing various output layers for action classification. Three proposed methods, i.e., the Symmetric Temporal Encoder (S-TE), Convolutional Temporal Encoder (C-TE), and Hierarchical Temporal Encoder(H-TE) were compared with data point feature classification and Principal Component Analysis (PCA). The study covered long-term predictions, which are more accurate with sliding windows. However, it needs further investigation concerning the parameter number and each network structure.

Pham The Hai et al. [41] developed a Human Action Recognition System (HARS) in both indoor and outdoor environments. Their approach involves observation sequences to train the Hidden Markov Model (HMM) and differentiate between seven activities. They extracted preferred features from a separate sequence of a human skeletal joint mapping, which involved Baum–Welch and the forward–backward algorithm to discover the optimal parameter of the individual HMM model. An automatic database update is needed to execute the system based on infrared light and time-of-flight technology from Kinect V2.

In different applications and domains, linear combination approaches across multiple levels of the hierarchy are suggested by Evangelos Spiliotis et al. [42]. Their strategy enables a more general non-linear combination of the reference prediction, which combines the accuracy and coherence of continued-to-optimize post-sample evidence-based forecasting, combined in a straightforward and appropriate model, without the need for comprehensive information for each series and level of the reconciled forecast, and using accuracy and bias as an evaluation method. The study was evaluated with Mean Absolute Scaled Error (MASE), Root Mean Squared Scaled Error (RMSSE), and Absolute Mean Scaled Error (AMSE). Correspondingly, the study enclosed several alternative paths (by focusing on cross-sectional hierarchical structures cases), did not generalize optimization objectives to each aggregation level, and did not maintain forecast uncertainty or computational complexity.

Alaa Sagheer et al. [43] developed the Deep Long Short-Term Memory (DLSTM) model in autoencoder. They applied transfer learning to mitigate the need to update each Long Short-Term Memory (LSTM) cell’s weight vector in hierarchical architectures and involve transfer learning methods to reduce the time-consuming and substantial quantity of data from various dimensions. The learned features are redistributed to the higher levels of the hierarchy for concurrent training to obtain more detailed conclusions and produce better forecasting models. The DLSTM was applied to publicly available datasets for Brazilian electrical power generation and Australian domestic tourism traveler nights. However, the study evaluated two criteria: forecasting accuracy and producing a coherent forecast but did not ensure forecast coherency at all hierarchy levels on a cross-temporal framework. The author also focused on the tourism and energy domains as examples.

Eduardo P. Costa et al. [44] is a machine learning technique used to address hierarchical classification problems in molecular biology related to proteins (organized hierarchically). They experimented with four hierarchical classification variant models (i.e., flat classification based on leaves, flat classification for all levels, top-down classification, and big-bang models) by using the G-protein-CouPled Receptor (GCPR) and enzyme protein families to predict protein functions by utilizing signature-generated protein sequences. The system requires a considerable level of detail for performance evaluation at the deepest classification level for each assessment, which is a system-automated procedure.

Yuliang Xiu et al. [45] developed a pose-tracking system for multi-person articulated poses by building associations between cross-frame and form poses. The system uses pose flows and non-maximum suppression to reduce redundant pose flows and re-link temporal disjoint. Nevertheless, the authors analyzed the proposed pose tracker’s short-term action recognition and scene understanding.

Hao-Shu Fang et al. [46] identified a subtle error in localization and recognition in the single-person pose estimator. By proposing a regional multi-person pose estimation framework, inaccurate human bounding boxes can be reduced. It has three-segment approaches and can deal with incorrect bounding boxes and redundant detections. Their research focused on human bounding box poses rather than the human detector and had difficulty distinguishing between overlapped people and similar objects when using human detection, resulting in undetected human poses.

Wojke et al. [47] proposed an approach to multi-object tracking. Their method integrates the appearance of simple online and real-time tracking. They applied offline pre-training and online application techniques to establish measurements to track associations using nearest neighbor queries in visual appearance spaces, which could track objects through more prolonged periods of occlusion. The study does not observe track jumping, which frequently occurs between false alarms.

Martinez et al. [48] proposed a method for predicting 3D positions from a 2D joint location. This system employs a simple deep feed-forward network to demonstrate the large portion of errors in a modern deep 3D pose estimation based on a visual analysis similar to a multi-layer perceptron. The approach needs more direct connections to visual evidence, which would also result in lower performance.

Hierarchical temporal memory has been linked to a Long Short-Term Memory (LSTM) network and cascade classifier [49,50,51]. There are principal distinctions, which is why LSTM ’gating’ is not proper because of the neuron model complexity of the HTM model, enforcing straightforward Hebbian learning rules locally and unsupervised, without utilizing back-propagation. Procedures in HTM are based on the activation and connectivity between neurons, utilizing binary units and weights.

Several HTM use cases in the research studies [52,53,54,55,56,57,58,59,60,61,62,63,64,65,66,67,68,69] and practices that have already been commercially tested include server anomaly detection using Grok [70], stock volume anomalies [71], rogue human behavior [72], natural language prediction [73], and geospatial tracking [74].

## 3. Problem Definition

This section clarifies some of the fundamental challenges behind hierarchical classification, including the motivations for evaluating significance or merit. It also explains the formal definition of classification, temporal classification, and hierarchical temporal classification to better comprehend this study. Furthermore, these detailed aspects make up the complete structure of an HTM network (as shown in Figure 2).

### 3.1. Classification

Classification challenges have already traditionally been considered as core components of deep learning, which would include supervised and semi-supervised learning problems.

A fundamental classification problem can be defined as producing an estimated label *y* from the *K*-dimensional of the input vector *x*, where:(1)$$x\in X\subseteq R^{{}_{K}}$$

For most of the common machine learning algorithms, the specific input variables must be of real value with the following: (2)$$y\in Y=\left\{C_{1},C_{2},...,C_{q}\right\}$$

This task employs the following classification rule or function:(3)g:X→Y
to predict new pattern labels [75,76].

### 3.2. Temporal Classification

For temporal classification terminology [77,78,79], there are several terms defined: channel, stream, and frame.

The channel formally can be defined as a function *c*, which maps from a set of timestamps *T* to a set of values, *V*, denoted as:(4)c:T→V
where *T* can be the set of times at which the value of channel *c* is defined, assume that:(5)$$T=\left[0,1,\cdots ,t_{max}\right]$$

The stream involves all of the values of the channels at once (collectively). A stream as a channel *s* sequence, for the same domain in each channel, is expressed as follows: (6)$$s=\left[c_{1},c_{2},\cdots ,c_{n}\right]\;s.t.\;\mathrm{domain}\left(c_{1}\right)=\mathrm{domain}\left(c_{2}\right)=\cdots =\mathrm{domain}\left(c_{n}\right)$$
where the number of channels in stream *s* can be noted as *n*.

The frame can be referred to as the values of all channels at a specific given point of time; formally, a frame can be defined as a function on an *s* stream and a *t* time. For a given stream, $s=[c_{1},c_{2},\cdots ,c_{n}]$, function fr is denoted as:(7)$$fr:\mathrm{domain}\left(c_{1}\right)\to \mathrm{range}\left(c_{1}\right)\times \mathrm{range}\left(c_{2}\right)\times \cdots \times \mathrm{range}\left(c_{n}\right)$$
(8)$$fr\left(s,t\right)=[c_{1}\left(T\right),c_{2}\left(T\right),\cdots ,c_{n}\left(T\right)]$$

For the temporal classification, let *S* be a training example set from a fixed distribution $D_{x\times z}$, input space $X=(R^{m})*$ as a set of all sequences in *m* dimensional real-valued vectors, and the target space x=l* involves all sequences over the alphabet *L* of labels.

Each example in *s* consists of a pair of sequences (x,z). The target sequence $y=(Y_{1},Y_{2},\cdots ,Y_{u})$ is, at most, as long as the input sequence of $x=(X_{1},x_{2},\cdots ,x_{T})$, so U≤T. The aim is to use *s* to train a temporal classifier g:X→Y [80].

### 3.3. Hierarchical Temporal Classification

As an example of the biologically inspired theories of machine intelligence, the Hierarchical Temporal Memory (HTM) theory was developed in 2008 by Jeff Hawkins [81,82]. The HTM Cortical Learning Algorithm (CLA), or simply, the HTM learning algorithm, is a type of HTM learning algorithm.

In [83,84], the authors look at the neocortex theory and its principles. HTM simulates the architecture and processes of the neocortex in the biological components of the brain. Comprehensive presentation of information about the main components of CLA, including the encoder, Spatial Pooler (SP) algorithm, and Temporal Memory (TM), which are classification and prediction algorithms, are listed below.

The hierarchical temporal classification task can be simplified as classifying a stream of data into a class hierarchy.

Stream classification can be formalized as learning instances from a stream *s* appearing incrementally as sequences of examples labeled as $\left\{x^{t},y^{t}\right\}$ for t=1,2,⋯,t, where *x* is a vector of attribute values and *y* is a class label $y\in \{K1,\cdots ,K_{l}\})$. The result in example $x^{t}$ is classified by classifier *C*, which predicts its class label [85,86].

A hierarchical classification task can be formulated as input X:s, where *s* can be denoted as the input stream. The category tree is denoted as *T* with the hierarchical layer as *L*. Categories as classes to be predicted are denoted by *Y*.

## 4. Materials and Methods

### 4.1. Materials

When conducting this research, the proposed framework for prediction probability was evaluated by considering two datasets. The dataset includes both a live validation experiment dataset and a farmer mocap dataset for practical evaluability assessment. The dataset analyzed is described as follows.

#### 4.1.1. Validation Dataset

Using accelerometer and gyroscope datasets obtained from [87], a large-scale egocentric vision dataset was created. The dataset consists of 20 daily activities with the following details: it is egocentric, non-scripted, uses a native environment, has 11.5 M frames, with 432 sequences, 39.596 action segments, 149 action classes, 454.255 object bounding boxes, 323 object classes, created by 32 participants, with 32 environments [88].

We utilized this dataset to calibrate and validate the gyroscope and accelerometer data, which were then implemented into quaternion calculations. The dataset representations are visualized in Figure 3.

Figure 3a shows the plot record for the accelerometer dataset. The blue color indicates the accelerometer coordinate for the X-axis. The green color indicates the accelerometer coordinate for the Z-axis, and the orange color indicates the accelerometer coordinate for Y-axis.

Figure 3b shows the plot record for the gyroscope dataset; the blue color indicates the gyroscope coordinate for the X-axis, the green color indicates the gyroscope coordinate for the Z-axis, and the orange color indicates the gyroscope coordinate for the Y-axis.

#### 4.1.2. Farming-Pack Mocap Dataset

This dataset [89] is the Farming-Pack mocap animation, as shown in Figure 4. It contains in total 24 mocap datasets with related farming activity source for Figure 4, e.g., working with a wheelbarrow (5 mocap datasets), picking fruit (3 mocap dataset), milking a cow (1 mocap dataset), watering a plant (1 mocap dataset), holding activity (4 mocap datasets), planting (1 mocap dataset), planting a tree (1 mocap dataset), pull out (2 mocap datasets), digging and plant seeds (1 mocap dataset), kneeling (1 mocap dataset), and working with a box (4 mocap datasets). Based on [90,91,92,93,94,95] for the sensor attachment, we evaluated the extracted rotation dataset for the rotation position.

### 4.2. Method

Classification is one of the tasks performed by machine learning. It is defined as the process of making predictions about the class or category of observed values or relevant data provided. In this investigation, information was collected from wearable devices that could read motion data and afterward perform a classification process depicted in Figure 5. Another tracking sensor developed by Sony, “mocopi [96]” uses a small light sensor that allows anyone to easily conduct full-body tracking in 3D. Vive [97] created another tracking sensor for full-body tracking.

Then, as a comparison, additional classification was performed to see how the outcome performances were compared to the mocap dataset.

Regarding local processing in a wearable device, we proposed several stages of the classification process, as shown in Figure 6.

#### 4.2.1. Feature Extraction

This section differentiates the visible components of the accelerometer and gyroscope to simplify the computation of input data interpretation. Two types of data received, i.e., the accelerometer and gyroscope dataset, were previously pre-processed by extracting quaternion [21,98,99,100] data from each dataset. Quaternion is a comprehensive way of describing a complex number; it was discovered by William Rowan Hamilton in the mid-19th century in 1843 [101]. The quaternion is a parameter representation of spatial rotation in three dimensions, extended to four dimensions. The representation of the quaternion calculation is described in Figure 7 [102], representing a wearable device in a smartwatch to read the accelerometer and gyroscope data.

The quaternion formula is denoted as follows [103]:(9)$$q=q_{w}+q_{x}i+q_{y}j+q_{z}k$$
(10)$$i_{2}=j_{2}=k_{2}=ijk=-1,ij=-ji=k,ki=-ik=j,jk=-kj=i$$

It is applied to *x*, *y*, *z* as follows, with θ being the rotation angle with vector *e* = ($e_{x}$, $e_{y}$, $e_{z}$).
(11)$$q=\left(\begin{array}{c} q_{w}\\ q_{x}\\ q_{y}\\ q_{z} \end{array}\right)=\left(\begin{array}{c} \cos\theta /2\\ e_{x}\sin\theta /2\\ e_{y}\sin\theta /2\\ e_{z}\sin\theta /2 \end{array}\right)$$
where *w* is denoted as the angle of rotation, and *x*, *y*, *z* are vectors representing the axes of rotation.

Quaternions are good choices for representing internal object rotations due to their efficiency in interpolation and single-orientation representation. However, the user interface presentation of quaternions is less intuitive compared to Euler angles, which are more familiar, intuitive, and predictable. Euler angles suffer from singularity and code-level issues, such as sequential rotation order storage, performance, and permutation support.

The matrix rotation contains nine elements with three rotational degrees of freedom, making this rotation matrix redundant. This matrix can illustrate the orientation of the reference frame, which is a more compact and intuitive way to define an orientation. Furthermore, the most convenient performance for reliable interpolation is represented by a quaternion.

#### 4.2.2. Encoding Data

Using a sparse pattern for input data on Hierarchical Temporal Memory (HTM), called Sparse Distributed Representation (SDR) [104], we used two types of encoders, i.e., *CategoryEncoder* and *RandomDistributedScalarEncoder*. *CategoryEncoder* is used to convert activity category values, as follows: [’P04’, ’P35’, ’P30’, ’P25’, ’P26’, ’P12’, ’P23’, ’P28’, ’P22’, ’P36’, ’P03’, ’P06’, ’P33’, ’P09’, ’P11’, ’P07’, ’P01’, ’P37’, ’P02’, ’P27’] for the first datasets (EPIC-KITCHEN-100 dataset). “P♯♯” (e.g., “P01”) is a participant or activity for each person. For the numbering of category data, we do not sort the data, but still use the category based on the dataset from [87,88]. For the Farming-Pack mocap dataset, *CategoryEncoder* is as follows: [M01, M02, M03, M04, M05, M06, M07, M08, M09, M10, M11], as shown in Table 1.

Moreover, for other data, for example, quaternions, we used the encoder, namely *RandomDistributedScalarEncoder* to encode numeric values in the form of floating-point values into an array of bits (in this case, sparse patterns).

#### 4.2.3. Spatial Pooling

The second step for the classifier is spatial pooling. In this step, we run the spatial pooler, which handles the column in a region and the input bit relation. The spatial pooler process returns a list of *activeColumns* columns.

#### 4.2.4. Temporal Memory

The next step for the classifier is sequence memory, temporal pooler, or temporal memory. In this step, we perform one step of the temporal memory algorithm for learning purposes, based on the computation of temporal memory to obtain the next time step prediction by identifying patterns that transition over time and recognizing spatial patterns across temporal sequences.

#### 4.2.5. Sparse Distributed Representation (SDR) Classifier

The classification algorithm used for the classification and prediction tasks in this experiment uses the SDR classifier. SDR is a data structure whose elements are binary, 1, or 0, representing the brain activity and, in the context of HTM, is a biologically-realistic model of neurons. The classifier receives input from the temporal memory algorithm’s SDR output, also known as the activation pattern. Furthermore, additional encoder information describes the target’s actual input. The SDR classifier performs this classification task similar to implementing a single-layered feedforward neural network.

There are three detailed stages in the SDR classification model: initialization, inference, and learning. All classes are initialized with zero values to equalize the probability values before learning in the initialization stage. In the next stage of inference, they calculate the probability distribution by applying the activation function at the activation level by performing calculations in the form of class prediction probabilities for each received input pattern. The final step of this classification is learning, which proportionally adjusts the weight to the gradient, calculates the error value for each output unit, and adds it to the appropriate element weight matrix.

#### 4.2.6. Performance Metrics Evaluation

The quaternion dataset as input data in this paper is a continuous value. In order to achieve reliability and validate the prediction of our suggested methodology, we applied related performance metrics. We used the following metrics to assess our prediction error rates and model the performance metrics: accuracy, precision, recall, *F*-score, Mean Absolute Error (MAE), Mean Squared Error (MSE), and Root Mean Squared Error (RMSE).

In classification problems, accuracy is used to calculate the percentage of correct predictions made by a model. In machine learning, the accuracy score is an evaluation metric that compares the number of correct predictions made by a model to the total number of predictions made. We calculate it by dividing the total number of predictions by the number of correct predictions.
(12)$$Accuracy=\frac{TP+TN}{TP+FP+TN+FN}$$

Precision is calculated as the percentage of correct predictions for a specific class out of all predictions made for that class.
(13)$$Precision=\frac{TP}{TP+FP}$$

In the case of binary classification, where we have an imbalanced classification problem, recall is calculated using the following equation:(14)$$Recall=\frac{TP}{TP+FN}$$
where True Positives (TP) are positive classes that are correctly predicted to be positive. False Positives (FP) are negative classes that are incorrectly predicted to be positive. True Negatives (TN) are negative classes that are predicted to be negative. False Negatives (FN) are positive classes that are incorrectly predicted to be negative.

The *F* -score (also referred to as the *F*1 score or *F*-measure) is a performance metric for machine learning models. It is a single score that combines precision and recall.
(15)$$F-score=2\times \frac{\left(precision\times recall\right)}{\left(precision+recall\right)}$$

*MAE* represents the difference between the initial input data and predicted results extracted by averaging the absolute difference over the dataset.
(16)$$MAE=\frac{1}{N}\sum_{i=1}^{N}\left|y_{i}-\hat{y}\right|\;$$

*MSE* represents the difference between the initial input data and predicted results extracted by squaring the average difference over the dataset, which can evaluate the degree of change in the data.
(17)$$MSE=\frac{1}{N}\sum_{i-1}^{N}{\left(y_{i}-\hat{y}\right)}^{2}$$

The error rate multiplied by the square root of the *MSE*, or the average size of the error, is denoted by *RMSE*. It is the square root of the average squared difference between the predicted and observed values.
(18)$$RMSE=\sqrt{MSE}=\sqrt{\frac{1}{N}\sum_{i-1}^{N}{\left(y_{i}-\hat{y}\right)}^{2}}$$
where $y_{i}$ is the observed value, $\hat{y}$ is the predicted value of $y_{i}$, $\bar{y}$ is the average of the observed value of $y_{i}$, and *N* is the number of samples.

## 5. Results

Data from the validation dataset and the Farming-Pack mocap dataset are time-series-based because they are sequential (temporal) or continuous data.

The first step of our framework is feature extraction. We extracted the accelerometer and gyroscope data from the validation dataset and convert it to quaternions based on these values. We also applied the same process to the Farming-Pack mocap dataset, where we extracted the position of the bones from the included (.bvh) file. The (.bvh) file is a text file that contains motion data. We converted the resulting positions to quaternions.

The quaternion results from the two datasets were then processed into the classifier in the following stages: data encoding, spatial pooling, temporal memory, and then classified using the Hierarchical Temporal Memory (HTM) classifier, and k-Nearest Neighbor (kNN) classifier.

At first, the HTM classifier did not learn the data patterns, resulting in poor performance. When learning runs, the classifier learns data patterns and makes reasonably accurate predictions. After learning for a while, the classifier can adapt to new patterns and make better predictions.

After running the experiment and achieving the results for specific data, we evaluated performance evaluation metrics in machine learning and statistics for this experiment. The evaluation performances of the HTM classifier with accuracy, precision, recall, F_Score, MSE, MAE, and RMSE are shown in Table 2. The rate of the performance metric is proportional to the percentage of the correct classification patterns applied to the dataset. We focused on solving hierarchical classification problems rather than specific algorithm problems due to the limited options for performance comparisons. We used the kNN algorithm in correlation with our validation and Farming-Pack mocap datasets.

The HTM can predict the data value in the next step based on previously stored patterns based on the structure of the HTM classifier system. When new data arrives and is processed, HTM compares it to the predicted value and calculates the difference.

Considering HTM as a continuous learning theory, there is no training, validation, or test set against the input data [105]. Data are learned and predicted continuously across all of the datasets.

## 6. Discussion

From our experiment, HTM can achieve 88.00% accuracy, precision of 0.99, recall of 0.04, and a F_Score of 0.09 for the validation dataset; it can achieve 54.00% accuracy, precision of 0.97, recall of 0.50, and a F_Score of 0.66 for the Farming-Pack mocap dataset.

Moreover, for the lower value of the average, we achieve an MSE = 5.10, MAE = 0.19, and RMSE = 0.38 for the validation dataset, and MSE = 0.06, MAE = 3.24, and RMSE = 1.51 for the Farming-Pack mocap dataset, which implies the higher performance of the prediction model.

When compared with other methods proposed in Table 3 and Table 4, our method achieves the following results: on the Human3.6M dataset, the RRNN method achieves an average of 0.97 and 0.23 for standard deviation, while CSS results in an average of 0.77 and 0.21 for standard deviation. The SkelNeT method achieves an average of 0.76 and 0.21 for standard deviation and our Skel-TNet method achieves an average of 0.73 and 0.21 for standard deviation. With the CMU mocap dataset, the CSS method results in an average of 0.61 and standard deviation of 0.18, the SkelNet method achieves an average of 0.60, and a standard deviation of 0.21, and our Skel-TNet method results in an average of 0.55 and standard deviation of 0.21.

The Carnegie Mellon University (CMU) mocap dataset was also evaluated by [40], who achieved classification rate results using various methods, i.e., data point = 0.76, PCA = 0.73, Deep Sparse Autoencoder (DSAE) = 0.72, S-TE = 0.78, C-TE = 0.78, and H-TE = 0.77. Another study by [41] used the HMM method with a training database and achieved accuracy results for activities such as stand = 88.67, walk = 86.20, run = 82.70, jump = 76.30, fall = 72.03, lie = 86.23, and sit = 92.70.

In the time-series dataset, [42] applied the xGBoost method on a tourism dataset and achieved MASE = 0.92, RMSSE = 1.16, and AMSE = 0.49. They also applied the random forest method and achieved MASE = 0.45, RMSSE = 0.67, and AMSE = 0.30. [43] applied the DLSTM method to the Brazilian electrical power generation dataset and achieved RMSE levels of 0 = 0.60, 1 = 1.02, and 2 = 2.91. They also applied the method to the Australian visitor nights or domestic tourism dataset and achieved RMSE levels of 0 = 2.27, 1 = 5.26, 2 = 6.34, and 3 = 8.07.

The study by [44] used hierarchical classification on two datasets. For the tourism dataset, they achieved the following accuracies: flat (leaves) = 61.33, flat (all levels) = 87.80, top-down = 87.80, and big-bang = 91.13. For the enzyme protein families dataset, they achieved the following accuracies: flat (leaves) = 62.73, flat (all levels) = 89.78, top-down = 89.78, and big-bang = 96.36.

Overall, this research represents the most dependable implementation with the optimum results. Wearable devices are used as IoT subsystem technology to collect motion sensor data, which are then processed locally and transferred to the cloud via the long-range (LoRa) network for decision-making.

Our statistical results confirm that our proposed method is both feasible and effective in addressing the constraints of time series datasets, and it is suitable for implementation in real rural farming environments for optimal solutions.

## 7. Conclusions

With the demand for farming activity monitoring in rural farming environments, this paper presents a classification prediction method for hierarchical temporal memory using the quaternion feature for farming safety activity monitoring. The obtained results support the proposed method’s ability to classify multiple activity classes. With 88.00% accuracy, precision of 0.99, recall of 0.04, F_Score of 0.09, MSE = 5.10, MAE = 0.19, and RMSE = 0.38 for the validation dataset, and 54.00% accuracy, precision of 0.97, recall of 0.50, F_Score of 0.66, MSE = 0.06, MAE = 3.24, and RMSE = 1.51 for the Farming-Pack mocap dataset, which implies the higher performance of a prediction model. Nevertheless, it would be exciting to look into the possibility of combining, extending, and evaluating other datasets and machine learning methods, which could be an interesting research direction. Examining computational complexity will be another future direction to maximize the proposed method further.

## Figures and Tables

**Figure 1 sensors-23-02951-f001:**
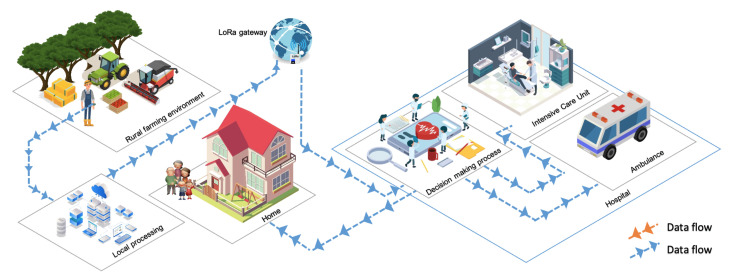
The Proposed farmer safety system in a smart farming environment.

**Figure 2 sensors-23-02951-f002:**
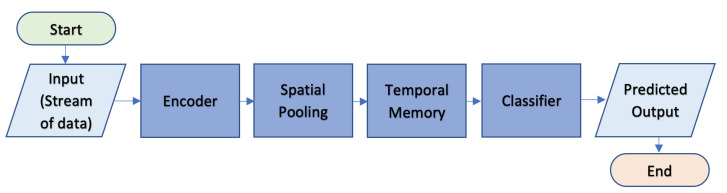
Hierarchical Temporal Memory (HTM) algorithm flowchart.

**Figure 3 sensors-23-02951-f003:**
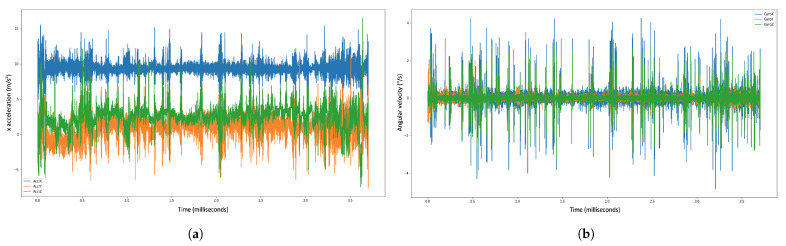
Example of the accelerometer and gyroscope data visualization for the validation dataset. AcclX, AcclY, and AcclZ denote the x−, y−, and z−axis of the 3D accelerometer, respectively. Correspondingly, GyroX, GyroY, and GyroZ denote the x−, y−, and z−axis of the 3D-gyroscope sensor, respectively. (**a**) Accelerometer dataset. (**b**) Gyroscope dataset.

**Figure 4 sensors-23-02951-f004:**
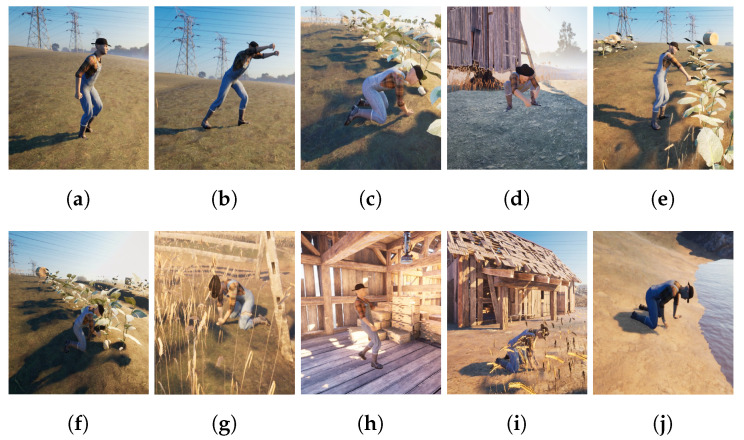
Farming-Pack mocap dataset [89]. (**a**) Wheelbarrow walk. (**b**) Wheelbarrow dump. (**c**) Picking fruit. (**d**) Milking cow. (**e**) Watering Plant. (**f**) Planting. (**g**) Pull plant. (**h**) Box walk. (**i**) Plant seeds. (**j**) Dig.

**Figure 5 sensors-23-02951-f005:**
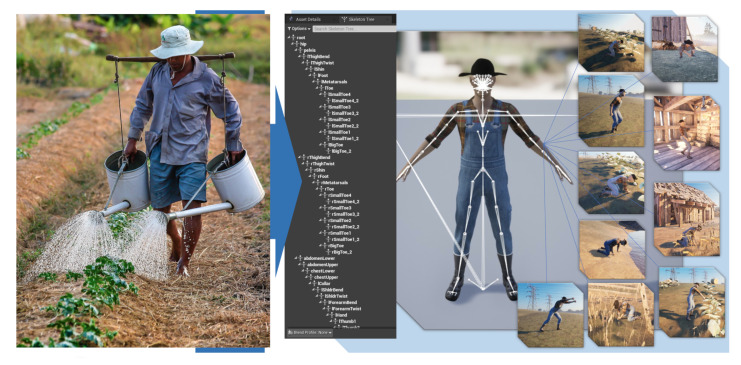
How to train the classifier.

**Figure 6 sensors-23-02951-f006:**
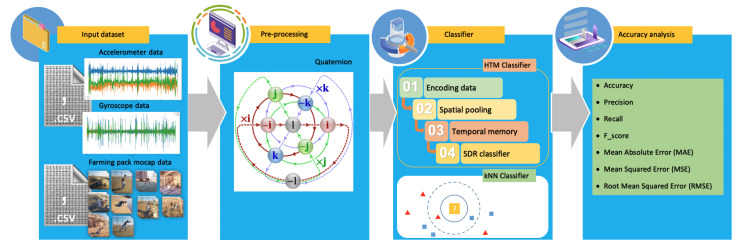
Proposed framework for prediction probability.

**Figure 7 sensors-23-02951-f007:**
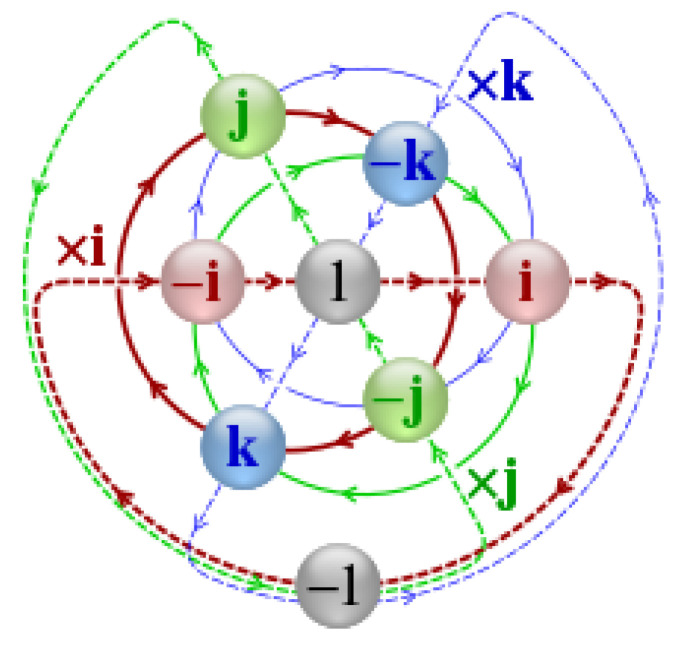
Quaternion visualization between axes (X, Y, Z) on a smartwatch-embedded gyroscope sensor and accelerometer sensor.

**Table 1 sensors-23-02951-t001:** *CategoryEncoder* for the Farming-Pack mocap dataset.

*CategoryEncoder*	Activity
M01	Working with a wheelbarrow
M02	picking fruit
M03	milking cow
M04	watering plant
M05	holding activity
M06	planting a plant
M07	planting a tree
M08	pull-out a plant
M09	digging and planting seeds
M10	kneeling
M11	working with a box

**Table 2 sensors-23-02951-t002:** Performance metrics evaluation for the validation dataset and Farming-Pack mocap dataset.

	HTM ^1^		kNN ^2^	
**Metrics**	**Validation Dataset**	**Farming-Pack Mocap Dataset**	**Validation Dataset**	**Farming-Pack Mocap Dataset**
Accuracy (%)	88.00	54.00	13.00	40.00
Precision	0.99	0.97	0.14	0.16
Recall	0.04	0.50	0.13	0.40
F_Score	0.09	0.66	0.12	0.23
MSE ^3^ (%)	5.10	0.06	30.00	25.00
MAE ^4^ (%)	0.19	3.24	13.00	3.39
RMSE ^5^ (%)	0.38	1.51	5.47	5.00

^1^ Hierarchical Temporal Memory (HTM); ^2^ k-Nearest Neighbor (kNN); ^3^ Mean Squared Error (MSE); ^4^ Mean Absolute Error (MAE); ^5^ Root Mean Squared Error (RMSE).

**Table 3 sensors-23-02951-t003:** The evaluation results of our proposed method compared with those of similar methods applied to the Mocap databases.

By	Dataset	Method	Results
[39]	Human3.6M		Average/standard deviation
		RRNN	0.97/0.23
		CSS	0.77/0.21
		SkelNeT	0.76/0.21
		Skel-TNet	0.73/0.21
	CMU mocap dataset		Average/standard deviation
		CSS	0.61/0.18
		SkelNeT	0.60/0.21
		Skel-TNet	0.55/0.21
[40]	CMU mocap dataset		Classification rate
		Data Point	0.76
		PCA	0.73
		DSAE	0.72
		S-TE	0.78
		C-TE	0.78
		H-TE	0.77
[41]	Training Database		Accuracy
		HMM	Stand = 88.67
			Walk = 86.20
			Run = 82.70
			Jump = 76.30
			Fall = 72.03
			Lie = 86.23
			Sit = 92.70

**Table 4 sensors-23-02951-t004:** Comparison evaluation results with a similar method applied on the time series and enzyme dataset.

By	Dataset	Method	Results
[42]	Tourism		Metrics
		xGBoost	MASE = 0.92
			RMSSE = 1.16
			AMSE = 0.49
	Sales		Metrics
		Random Forest	MASE = 0.45
			RMSSE = 0.67
			AMSE = 0.30
[43]	Brazilian electrical	DLSTM	Metrics: RMSE
	power generation		level 0 = 0.60
			level 1 = 1.02
			level 2 = 2.91
	Australian visitor nights	DLSTM	Metrics: RMSE
	of domestic tourism		level 0 = 2.27
			level 1 = 5.26
			level 2 = 6.34
			level 3 = 8.07
[44]	Tourism		Metrics:accuracy
		Flat (Leaves)	61.33
		Flat (all levels)	87.80
		Top-Down	87.80
		Big-Bang	91.13
	Enzyme protein families		Metrics:accuracy
		Flat (leaves)	82.73
		Flat (all levels)	89.78
		Top-Down	89.78
		Big-Bang	96.36

## Data Availability

Not applicable.

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
