# Peer review of "IoT and Deep Learning-Based Farmer Safety System"

_sensors, 2023, doi:10.3390/s23062951_

Round 1

Reviewer 1 Report

Paper deals with important task. The authors investigated the validation and simulation dataset to determine whether accidents occur to farmers

1. Abstract should be extended using details of the proposed approach

2. It would be good to clearly show the main contribution of this paper in the Introduction sections

3. I abstract I found that the authors solve classification ttask. If this is correct please used performance indicators like F1 and others to evaluate your results. In the Resuls section I fount only indicators that are typicaly used in regression problem

4. Comparison is strange. Did you implemented other methods for comparison with the same train/tast data or you just take results from existing papers

5. Conclusions should be extended using numerical results obtained by authors of this paper. 

Author Response

Manuscript ID: sensors-2250244   Respected Editor, Dr. Rongxing Lu, Dr. Sajjad Dadkhah, Dr. Jianting Ning, and Dr. Beibei Li. Sensors (ISSN 1424-8220), Section "Sensor Networks" Special Issue "Security and Privacy of the Internet of Things for Industrial Applications"     Thank you for allowing us to submit a revised draft of the manuscript:   Title :  IoT and Deep Learning Based Farmer Safety System Authors : Yudhi Adhitya * , Grathya Sri Mulyani , Mario Köppen , Jenq-Shiou Leu   We appreciate the time and effort the Editorial Board and reviewers placed into providing feedback on our manuscript. We are grateful for the insightful comments that will help us improve our paper. We have taken into account all of the suggestions made by reviewers. Please see the section below for a detailed response to the reviewer's comments and concerns.

Reviewer 2 Report

1. In the abstract (line 10-13) authors of this manuscript claim that "The computational framework with an application of wearable device technology connected to ubiquitous systems; and statistical results verify that our proposed method is feasible and effective in solving the problem’s constraints in a time series data set that is acceptable and usable in a real rural farming environment for optimal solutions". However, it is not clear in the text of manuscript. 

2. Literature review is weak. 

3. Specify how the wearable devices connect to the proposed solution. which method is applied for connect in a real time.

4. Use any standard method to present methodology.

5. why do you present figure 2 in this manuscript? Is it needed?

6. Please discuss your findings? It is not clear what you are going to present and how the results contribute in ongoing research in the same research area.

7. Add more latest references from 2022-23.

Author Response

(The authors gave the same response as above.)

Reviewer 3 Report

Comments to Author

The authors proposed a quaternion to represent 3D rotation as an input feature. They exploited quaternion as four-dimensional data as a feed input for Hierarchical Temporal Memory (HTM) classifier. The idea of the paper is good; however, my comment to authors as follows:

1) The authors claim that they used IoT to build the former safety system, however, I did not see how and where they used the IoT in their proposed system.

2)  Abstract should clearly mention the proposed system including the IoT.

3) Authors did not present the network part of how IoT devices will make the former system is safe.

4) The contributions should be clearly highlighted before/after section 2.    

5) I recommend adding flowchart or data flow diagram, and an algorithm at the end of section 3.

6) In all results discussion, the authors should explain why the proposed system prediction method provides accurate result. Justification of all results is required for each diagram/table.

7) Conclusion should be enhanced to summarize the proposed method, achievement, and future work in one paragraph.  

Author Response

(The authors gave the same response as above.)

Round 2

Reviewer 1 Report

The F1 score and total accuracy is very low but maybe this ask is very specific

The paper can be accepted in current form.

Reviewer 2 Report

The authors have revised this manuscript according to my previous comments. I am agree to accept this manuscript in its present form.

Reviewer 3 Report

Authors addressed all my comments.